

# Estimation of anthropogenic and volcanic SO₂ emissions from satellite data in the presence of snow/ice on the ground

Vitali E. Fioletov[1], Chris A. McLinden[1], Debora Griffin[1], Nickolay Krotkov[2], Can Li[2,3], Joanna Joiner[2], Nicolas Theys[4], and Simon Carn[5]

[1]Air Quality Research Division, Environment and Climate Change Canada, Toronto, ON, Canada
[2]Atmospheric Chemistry and Dynamics Laboratory, NASA Goddard Space Flight Center, Greenbelt, MD, USA
[3]Earth System Science Interdisciplinary Center, University of Maryland, College Park, MD, USA
[4]Royal Belgian Institute for Space Aeronomy (BIRA-IASB), Brussels, Belgium
[5]Department of Geological and Mining Engineering and Sciences, Michigan Technological University, Houghton, MI 49931, USA

Correspondence to: Vitali.Fioletov@outlook.com or Vitali.Fioletov@ec.gc.ca

**Abstract.** Early versions of satellite nadir-viewing UV SO₂ data products did not explicitly account for the effects of snow/ice on retrievals. Snow covered terrain, with its high reflectance in the UV, typically enhances satellite sensitivity to boundary layer pollution. However, a significant fraction of high-quality cloud-free measurements over snow is currently excluded from analyses. This leads to increased uncertainties of satellite emissions estimates and potential seasonal biases due to the lack of data in winter months for some high-latitudinal sources. In this study, we investigated how OMI and TROPOMI satellite SO₂ measurements over snow-covered surfaces can be used to improve the annual emissions reported in our SO₂ emissions catalogue (version 2, Fioletov et al., 2023). Only 100 out of 759 sources listed in the catalogue have 10% or more of the observations over snow. However, for 40 high-latitude sources, more than 30% of measurements suitable for emission calculations were made over snow-covered surfaces. For example, in the case of Norilsk, the world's largest SO₂ point source, annual emissions estimates in the SO₂ catalogue were based only on 3-4 summer months, while addition of data for snow conditions extends that period to 7 months. Emissions in the SO₂ catalogue were based on satellite measurements of SO₂ slant column densities (SCDs) that were converted to vertical column densities (VCDs) using site-specific clear-sky air mass factors (AMFs), calculated for snow-free conditions. The same approach was applied to measurements with snow on the ground whereby a new set of constant, site-specific, clear sky with snow AMFs was created, and these were applied to the measured SCDs. Annual emissions were then estimated for each source considering (i) only clear-sky snow-free days, (ii) only clear-sky with snow days and (iii) a merged dataset (snow and snow-free conditions). For individual sources, the difference between emissions estimated for snow and snow-free conditions is within ±20% for three quarters of smelters and oil and gas sources, and with practically no systematic bias. This is excellent consistency given that there is typically a factor of 3-5 difference between AMFs for snow and snow-free conditions. For coal-fired power plants, however, emissions estimated for snow conditions are on average 25% higher than for snow-free conditions; this difference is likely real and due to larger production (consumption of coal) and emissions in wintertime.



# 1    Introduction

Sulfur dioxide ($SO_2$) is a major air pollutant that plays an important role in atmospheric chemistry, contributes to aerosol formation, adversely affects the environment and human health, and impacts climate (Fischer et al., 2019; Hansell and Oppenheimer, 2010; Longo et al., 2010; Robock, 2000). $SO_2$ also leads to acid deposition that affects terrestrial ecosystems (Dentener et al., 2006; Hutchinson and Whitby, 1977; Vet et al., 2014; Fedkin et al., 2018). Most of atmospheric $SO_2$ is a result of human activity and the major sources of anthropogenic emissions are coal-burning power plants, oil refineries and smelters (Klimont et al., 2013; Smith et al., 2011), while volcanoes are the primary natural source of $SO_2$ (Carn et al., 2017; Oppenheimer et al., 2011). A proper account for $SO_2$ content in the atmosphere is necessary for climate, weather, and air quality models (e.g., Liu et al., 2018; Stenchikov et al., 2021; Ukhov et al., 2020). Such models require information about $SO_2$ emissions from anthropogenic and natural (mostly volcanic) sources.

The history of satellite $SO_2$ measurements in the Earth's atmosphere goes back to 1982, when volcanic $SO_2$ from the El Chichón eruption in 1982 was first retrieved from measurements of the Total Ozone Mapping Spectrometer (TOMS) and the Solar Backscattered Ultraviolet (SBUV) instruments on the Nimbus 7 satellite (Krueger, 1983; McPeters et al., 1984). Over the last two decades, several satellite UV–visible instruments have been used for the monitoring of atmospheric $SO_2$. They include the Global Ozone Monitoring Experiment (GOME) in 1995-2011 (Eisinger and Burrows, 1998; Khokhar et al., 2008); the SCanning Imaging Absorption spectroMeter for Atmospheric CHartographY (SCIAMACHY) in 2002-2012 (Bovensmann et al., 1999; Lee et al., 2009); the Global Ozone Monitoring Experiment-2 (GOME-2) instrument, 2006-present, (Bramstedt et al., 2004; Callies et al., 2000; Nowlan et al., 2011; Rix et al., 2012), and the Ozone Mapping and Profiler Suite (OMPS), 2011- present (Zhang et al., 2017). However, the spatial resolution of these instruments was not very high, above about 40 km, that made it difficult to monitor small emission sources.

Satellite $SO_2$ measurements with high spatial resolution that is sufficient to detect hundreds of $SO_2$ emissions sources started with the launch of the Dutch–Finnish Ozone Monitoring Instrument (OMI) (Levelt et al., 2018, 2006) on NASA's Earth Observing System (EOS) –Aura spacecraft (Schoeberl et al., 2006) in 2004. OMI has the spatial resolution as fine as $13 \times 24$ km² at nadir and is able to provide daily, nearly global maps of $SO_2$ VCDs (Krotkov et al., 2016; Li et al., 2020). The TROPOspheric Monitoring Instrument (TROPOMI) (Theys et al., 2017) on the ESA Copernicus Sentinel-5 Precursor (S-5P) spacecraft (Veefkind et al., 2012) launched in 2017 has even higher spatial resolution, $5.5 \times 3.5$ km² and also is able to provide daily global coverage.

Satellite measurements of $SO_2$ are available in the form of vertical column density (VCD) that represent the total number of molecules in a vertical column per a unit of area. VCDs and $SO_2$ emissions estimated from them are used for multiple applications such as evaluation of long-term changes and trends on a regional and global scale in $SO_2$ VCDs (Krotkov et al., 2016) and surface concentrations (Kharol et al., 2017) and for assessment the efficiency of industrial clean technology solutions in reducing of $SO_2$ emissions (Ialongo et al., 2018; McLinden et al., 2020). Such measurements are also an important source of information about volcanic $SO_2$ (Carn et al., 2013; Inness et al., 2022; Krueger et al., 1995; McCormick et al., 2013)



and volcanic $SO_2$ is is used to estimate volcanic carbon dioxide ($CO_2$) fluxes (Fischer et al., 2019). Satellite $SO_2$ VCDs are also used to estimate emissions from anthropogenic $SO_2$ sources (Fioletov et al., 2013, 2015; McLinden et al., 2012, 2020; Qu et al., 2019) and to update the available emissions inventories (Liu et al., 2018; Ukhov et al., 2020).

A global catalogue of large $SO_2$ sources (hereafter referred to as the "$SO_2$ catalogue") and their emissions estimated from satellite $SO_2$ VCD observations was developed (Fioletov et al., 2016; McLinden et al., 2016). The $SO_2$ catalogue has been updated to version 2 recently (Fioletov et al., 2023). This version 2 covers the 2005-2022 period and contains information on a total of 759 point sources emitting from about 10 kt yr$^{-1}$ to more than 4000 kt yr$^{-1}$ of $SO_2$. The $SO_2$ catalogue also includes information about the country and source type, so data can be grouped and summed by them. The $SO_2$ catalogue contains annual emission estimates for 106 volcanoes, 477 power plants, 74 smelters, and 102 sources related to the oil and gas industry. Satellite data from version 2 of OMI and OMPS as well as from a recent version of the $SO_2$ TROPOMI dataset were used to estimate emissions for the version 2 $SO_2$ catalogue.

Early versions of satellite $SO_2$ data products were based on retrieval algorithms that estimated $SO_2$ slant column density (SCD) first and then converted it to VCD using a single conversion factor that corresponds to typical snow-free surface conditions (Khokhar et al., 2005; Lee et al., 2008; Thomas et al., 2005). For example, in the original OMI algorithm, all SCD values were then converted to planetary boundary layer (PBL) VCDs by applying a constant AMF = 0.36 that was appropriate for anthropogenic pollution in the eastern US in summer (Krotkov et al., 2008, 2006; Li et al., 2013). Subsequently, PBL AMFs were calculated using a climatological $SO_2$ profile and information about surface reflectivity, solar zenith angle, and other characteristics (Lee et al., 2009; Li et al., 2020c; Theys et al., 2017). For example, Lee et al. (2009) used shapes of $SO_2$ profiles determined from a global 3-D model of tropospheric chemistry (GEOS-Chem) along with a surface reflectivity climatology, and information about cloud cover and aerosols. McLinden et al. (2014) calculated site-specific AMFs based on information about the local climatological surface reflectivity, aerosols, and boundary layer height assuming that all $SO_2$ is in the boundary layer. Based on that approach, estimated emissions reported in the Versions 1 and 2 $SO_2$ catalogues were calculated using a single site-specific AMF that corresponds to measurements under snow-free conditions only (Fioletov et al., 2016, 2023). Thus, potentially high-quality measurements collected under snow conditions were excluded from the emission estimates increasing the uncertainties of the estimates and introducing potential biases due to the lack of data in winter months for some high-latitudinal sources.

More recently, NASA version 2 (V2) of the principal component analysis (PCA) algorithm for OMI/OMPS data processing used updated a priori $SO_2$ vertical profiles that were calculated using the Goddard Earth Observing System Version 5 (GEOS-5) global model and the corresponding OMI pixel-specific AMFs (Li et al., 2020). Thus, V2 $SO_2$ data included VCDs calculated over snow and ice using OMI-measured Lambertian Equivalent scene Reflectivity (LER) and are in a better agreement with summertime values (Li et al., 2020). In this study, we consider two data sets of satellite $SO_2$ measurements taken under snow on the ground conditions: (1) the NASA OMI V2 $SO_2$ VCD dataset and (2) the dataset based on OMI and TROPOMI SCDs converted to VCDs using two site-specific time-independent AMF values calculated for each source: one for conditions with snow and another for conditions without snow on the ground. Both data sets are used to calculate $SO_2$ point





source emissions in order to improve the existing global catalogue of large SO$_2$ sources and their emissions (Fioletov et al., 2023). The study is focused on OMI data analysis because of its 18-year long record of observations with spatial resolution suitable for emissions estimates for hundreds of sources. TROPOMI data have a superior spatial resolution, but a much shorter record and were mostly used for some illustrations. We did not use OMPS data in this study because the number of measurements is not large enough to reliably estimate emissions under snow conditions for most of the sources.

This article is organized as follows: the data sets and AMF values are described in **Section 2**. **Section 3** discusses the differences between emissions estimated for conditions with and without snow on the ground and provides an overview of the estimated emissions using all data (with and without snow on the ground). **Section 4** summarizes the findings of the study.

## 2        Data sets and emissions estimates

The OMI and TROPOMI Level 2 SCD data sets, data selection criteria, and the emission calculation algorithm used in this 110 study are identical to those in Fioletov et al. (2023) and we just briefly review them here. However, we included measurements taken in presence of snow on the ground that were previously excluded from the emissions calculations. The estimated annual emission rates are given in metric kilotonnes of SO$_2$ per year (kt y$^{-1}$) and the VCD values are given in Dobson Units (DU, 1 DU = 2.69•10$^{16}$ molec•cm$^{-2}$).

### 2.1 OMI data

OMI was launched on NASA's Earth Observing System (EOS - Chemistry) Aura satellite on 15 July 2004 (Schoeberl et al., 2006). Aura is on a sun-synchronous polar orbit that crosses the equator at about 13:45 local mean time. The OMI spatial resolution is 13 km × 24 km at nadir and up to 30 km × 100 km at the swath edges (de Graaf et al., 2016). The OMI wide swath (~2600 km) is divided into 60 cross-track positions; the first and last 10 cross-track positions were excluded from our analysis to limit the across-track pixel width from 24 km to about 40 km. About half of OMI pixels have been affected by a 120 field-of-view blockage and stray light (the so-called "row anomaly") since 2007 (Levelt et al., 2018), and they were also excluded from our analysis.

In this study, we used the V2 OMI PCA SO$_2$ (OMSO2 V2) data (Li et al., 2020) aimed at anthropogenic SO$_2$ pollution for the period 2005-2022. Only measurements with solar zenith angles (SZA) less than 70° are used. In the V2, a priori SO$_2$ vertical profiles are based on average GEOS-5 global model simulations (72 vertical layers, 0.5° latitude by 0.667° longitude 125 resolution) for the period 2004–2014. Monthly climatological profiles are calculated for each grid cell and then used as a priori profiles in the SO$_2$ VCD retrievals. There are two potential issues in this approach. First, the period may not reflect the present emission levels as SO$_2$ emission sources, and their emission levels are changing over time. Second, such climatological profiles represent "average" conditions within the model grid cell, that could be different from plume profiles near pollution sources (i.e., the signals near the emissions sources are diluted by the relatively coarse spatial resolution as compared with the 130 resolution of OMI pixels). While such profiles near the source assume high concentrations in the boundary layer, if we move





away from the source, the model simulated SO$_2$ profile is more heavily weighted towards the free troposphere due to less emissions in the boundary layer and chemical loss and deposition in the boundary layer. As a result, the Air Mass Factors (AMF = SCD/VCD) depend on the distance from the source: they are lower over the source and getting larger further away from it. Both of these issues affect the emissions estimates based on OMSO2 V2 VCD data.

Another data set of VCDs used for emission estimates in this study is also based on OMSO2 V2 data but uses a different approach to calculate VCDs. OMSO2 V2 also provides SO$_2$ SCDs produced by spectral fitting using SO$_2$ absorption cross sections. In the SO$_2$ catalogue, these SCDs were converted to VCD using a constant over time site-specific air mass factor (AMF$_{source}$ =SCD/VCD) calculated at Environment and Climate Change Canada (ECCC) as discussed in Section 2.4 (McLinden et al., 2014). The same approach is used here; however, two AMFs are calculated for each source: for snow and

snow-free conditions. We will refer to these AMFs as to "ECCC AMFs".

     A 10% empirical correction is applied to OMI VCDs for the version 2 SO$_2$ catalogue (Fioletov et al., 2023). This is done to match the values from the original SO$_2$ catalogue, that were validated against available emission inventories. This is a relatively small correction compared to uncertainties in AMF and lifetime (18% and 35%, respectively (Fioletov et al., 2023)).

## 2.2 TROPOMI data

While this study is largely focused on OMI data, we also use TROPOMI measurements for some illustrations and comparisons. TROPOMI was launched on the ESA Copernicus Sentinel 5-Precoursor satellite on October 13, 2017 (Veefkind et al., 2012). The instrument contains a UV spectrometer that is used to retrieve SO$_2$. The TROPOMI detector has 450 cross-track positions. We excluded the first and last 20 of them since their measurements have a relatively high noise level (Fioletov et al., 2020). The TROPOMI spatial resolution for the centre of the swath was approximately 7 km $\times$ 3.5 km until August 6, 2019 when it

was improved to 5.5 km $\times$ 3.5 km.

     Here, we use TROPOMI data processed by the Covariance-Based Retrieval Algorithm (COBRA) (Theys et al., 2021) for the period from April 1, 2018, to December 31, 2022. Similar to the OMI analysis, only data with SZA<70° are used for the study. Note that TROPOMI and OMI SO$_2$ SCD retrieval algorithms use different SO$_2$ cross-section temperature (203K and 293K respectively). As a result, reported TROPOMI SCDs are 22% lower than they would be retrieved using cross-section for

293K (see Theys et al., 2017, their Figure 6). For this reason, we increase TROPOMI SCDs used this study by 22% (see also Fioletov et al., 2020; Theys et al., 2017). For TROPOMI, the same conversion of SCD to VCD with the same site-specific ECCC AMFs as for OMI is used.

## 2.3 Emissions estimates

The emission calculation algorithm was identical to that used to calculate emissions for the SO$_2$ catalogue (Fioletov et al.,

2015; 2016; 2023). It is based on the fitting algorithm to estimate the average total SO$_2$ mass at and downwind of the source in order to derive emissions, assuming a constant lifetime. The pixel-based SO$_2$ VCDs were fit to a two-dimensional exponentially-modified Gaussian plume dispersion function (Fioletov et al., 2016) that depends on the wind speed and





direction at the pixel location on the time of satellite overpass. The parameter of the fit represents the total SO$_2$ mass, emitted from the source. As we are interested in annual emissions, the fitting was done using all pixels near the source collected over

one year. The exact fitting area depends on the wind speed and emission strength as described by Fioletov et al. (2016). The wind speed and direction are obtained from most recent ERA5 reanalysis wind data (C3S, 2017). A recent study shows that ERA5 data are reliable and have practically no bias on offshore and flat onshore locations, although wind uncertainties are greater at mountainous and coastal sites (Gualtieri, 2022). Available from ERA5, U- and V- (west-east and south-north, respectively) wind component data are grouped into 1 km-thick layers and the mean wind speed and direction are calculated

for each level as done for the original catalogue. The winds for the layer that corresponds to the height of a source are used by the algorithm.

As in the SO$_2$ catalogue, the fitting function depends on two prescribed non-linear parameters: the plume width and SO$_2$ effective lifetime (or decay time, τ). The plume width is related to the size of satellite pixels and the spatial extent of the source plume, and it should not depend on the presence of snow. The same values as in the SO$_2$ catalogue were used for the

plume width (20 km for OMI and 15 km for TROPOMI). We also used the same value of τ=6 hours for the lifetime for both snow and snow-free conditions for all sources. The lifetime uncertainty is one of the major contributors to the overall uncertainty of the emission estimates. For snow-free conditions (i.e., for the SO$_2$ catalogue), it was estimated that for 80% of all sources τ is between 3 and 9.5 h with the mean of about 6 h (Fioletov et al., 2016). If τ is included as one of the fitting parameters, owing to its non-linear nature, the emission uncertainties may become too large, particularly for small sources. A

single prescribed τ value reduced random uncertainties of emission estimates, although it potentially introduces a multiplicative bias in estimated values for some sources (Fioletov et al., 2016). For snow conditions, we tested the same value of τ=6 hours and found that, on average, the estimated emissions have only a small (3-4%) bias compared to these for snow-free conditions (see also section 3.3). A single value for snow and snow-free conditions also substantially simplifies the algorithms because otherwise it is not clear how to treat days with partial snow cover around the source.

As part of the OMI level 1B to level 2 processing, before the SO$_2$ retrieval, cloud radiance fraction (CRF), cloud optical centroid pressure, and effective scene pressure are derived using the OMCLDRR algorithm (Joiner and Vasilkov, 2006; Vasilkov et al., 2008). This information is essential to correctly interpret the SO$_2$ SCDs and to calculate the AMF. OMI itself cannot discriminate between snow and clouds as both act as strong reflectors of UV/visible light. Hence, the OMI cloud algorithm uses external snow/ice flag from Near-real-time Ice and Snow Extent (NISE) dataset. Misdiagnosing the presence

of snow, and the reflectivity of snow if it is present, can lead to errors in the cloud fraction and dependent products (see e.g., Vasilkov et al., 2010).

Emissions estimates were done for pixels with and without snow on the ground under "clear sky" conditions. For snow-free pixels, clear sky conditions were defined as those with CRF of less than 0.3. For snow-covered pixels, the clear sky conditions were determined from comparisons of terrain and effective scene pressures (ESP). In OMI version 2 data, the ESP

from the OMCLDRR algorithm is compared with the terrain pressure. If the difference is within 50 hPa, the pixels was considered as cloud-free and CRF was set to zero (Li et al., 2020). If it is greater that 100 hPa, then the pixel was likely cloudy,



and CRF was set to one. For pixels having scene and terrain pressure differences between 50 and 100 hPa, "unambiguous cloud detection is not possible" (Li et al., 2020), but CRF was set to 0. We tested both the 50 and 100 hPa cut-off limits for determination of the clear sky conditions for several sources. It appears that there is no systematic difference in estimated
emissions between the two limits, however the number of clear sky pixels is 20%-40% larger for the 100 hPa limit than for the 50 hPa limit making the emission uncertainties smaller. For this reason, we used the 100 hPa limit in this study.

Similarly, TROPOMI COBRA data contain information about the terrain and cloud top pressure taken from the TROPOMI cloud product (Loyola et al., 2018). In case of TROPOMI, we also applied the condition that the difference between the terrain and cloud top pressure should be less than 100 hPa. It should be noted that OMI scene pressure may be different
from the TROPOMI cloud top pressure data product, however the comparison of these two data products is outside the scope of this paper.

**2.4. Site-specific AMFs**

As in the previous studies (Fioletov et al., 2016; McLinden et al., 2016), pre-calculated site-specific and time-independent AMF values were used to calculate the SO$_2$ VCDs and emissions for each source location. Here, two different AMF values are calculated for each emission source: one for snow-free conditions (as in the previous publications) and another for the
snow on the ground conditions. The AMF calculation method is based on a general approach from McLinden et al. (2014) modified as described in Fioletov et al. (2022). The AMFs were calculated for a subsample of OMI observations within 100 km of the emission source coordinates. Every 100$^{th}$ observation from every 3$^{rd}$ year were used that yields several thousand observations and is sufficient to represent conditions (cloud fraction, viewing and solar geometry, and seasonal sampling) for a given source. As before, AMFs were calculated using the MODIS MCD43C3 (v6.1) reflectance data (Schaaf et al., 2002).
MODIS LER (43C3) albedo at 477 nm is used, with albedo calculated separately for snow and snow-free conditions. For snow-free conditions, an annually varying monthly mean was used considering only snow free MODIS pixels and the MODIS 477 nm albedo is mapped to SO$_2$ wavelengths using OMI 0.5 degree climatology at ratio of 342 nm to 477 nm albedos (see McLinden et al., 2014, sec 3.2.2). For snow conditions, the 477 nm albedo is used as is.

For snow conditions, a single snow albedo map was created by considering all years and months using pixels with a snow flag of 1. The final, site-specific AMFs were the average over these individual AMFs, separating them into snow and snow-free categories. Note that the SO$_2$ profile is estimated based on the elevation of the source and a climatological boundary-layer height (as a function of latitude, longitude, month, and UTC hour) from von Engeln and Teixeira, (2013), assuming a constant mixing ratio in the boundary layer and zero elsewhere.

The snow cover information was obtained from the Interactive Multisensor Snow and Ice Mapping System measurements (IMS) (Helfrich et al., 2007) which has been shown to be one of the best for identifying snow on the ground (Cooper et al. 2018). While the OMI datafiles contain information about snow/ice obtained from the NISE database, this product is known to frequently misidentify snow as no-snow when the layer is thin (McLinden et al., 2014; Cooper et al.,





2018). The use of a single, consistent product is preferred to avoid possible sampling differences when data form other
instruments are used for emission calculations for the merged SO₂ catalogue.

Although the NASA VCD retrievals do not explicitly calculate AMFs, we obtained an effective AMF by taking the
ratio of SCD to VCD in order to better understand the differences between emissions estimates from the two data sets. "NASA
AMFs" were calculated for each source as a median value of the SCD to VCD ratios for all pixels within a 25 km radius around
the source. As mentioned in Section 2.1, the NASA AMFs depend on the distance from the source: they are lower over the
source and getting larger further away from it. We used a small area around the source with a 25 km radius to have a better
representation for AMF in the plume. Further reduction of the radius does not change the AMF value but increases the estimate
uncertainty. For the ECCC AMFs, the profiles are determined by the PBL height and therefore is nearly constant within 200-
300 km around an emission source used for satellite data fitting.

## 3. Results

### 3.1 Source statistics

At present, there are 759 sources of emissions listed in the SO₂ catalogue (Fioletov et al., 2023), but many of them are located
at low latitudes where snow conditions are very rare. **Figure 1** shows a map of the sources where OMI measurements are
available, in the presence of snow on the ground. The size of the dots on the map is proportional to the fraction of measurements
under snow conditions among all measurements suitable for emission estimates. Most of the industrial sources are in Russia,
Canada, Kazakhstan, and a region in north-west China. There are also several volcanic sources in Alaska and Kamchatka. The
largest fraction, about 60%, is at four sites in norther Russia including Norilsk, the world's largest anthropogenic SO₂ point
source. The fraction is less than 10% for most of the US and European sources. For a total of 100 sources the fraction is more
than 10%, and we focus this study on the analysis of data from these sources. Note that the utilized snow cover information
from IMS does not cover the southern hemisphere. This is not a problem since sources in the southern hemisphere do not
normally have snow on the ground. The only exceptions are the Erebus volcano in Antarctica and the South Sandwich Island
volcanoes (Michael and Montagu), but they are always covered by snow. There are also several volcanic sources in the northern
and southern latitudes located on small islands that we did not consider in this study: while the volcanoes themselves may have
snow cover, the islands are small and surrounded by open ocean.

As snow reflects more light than snow-free surfaces, the SCD values over snow are higher, and VCD values are
overestimated if AMF for a snow-free albedo is used to convert SCD to VCD. This overestimation is illustrated in **Figure 2**
that shows the 2005-2022 mean SO₂ SCDs and VCDs for snow and snow free conditions for Norilsk. SCDs for snow
conditions are about 7 times greater than those for snow-free conditions; however, VCDs are very similar.

The ratio of AMF for snow conditions to AMF for snow-free conditions is calculated to estimate potential biases if
measurements over snow are used to calculate VCD using a snow-free AMF. If the ratio is close to 1, then there is no need to
introduce an additional AMF for snow conditions. **Figure 3** shows the map of such ratios calculated for ECCC AMFs. The





ratios are higher over the plains in the US and Kazakhstan as well as over Canadian and Russian tundra and lower over boreal forests in Alaska, Canada, and Russia. The results for NASA AMFs are similar (not shown), although there are also some differences. The absolute values of the ratios are typically higher for ECCC AMFs (the range is from 1 to 8) than for NASA AMFs (the range is from 1 to 6). Note that NASA AMFs the NISE snow/ice flag is known to miss some snow. There are also

two sources in northern Russia (marked by a red ellipse), where the difference between NASA and ECCC ratios is particularly large. These sources came online after 2012, and their emissions are likely not included in the GEOS-5 model simulations from which the a priori vertical profile are derived for the NASA OMSO2 algorithm. Therefore, modelled profiles used in the standard OMSO2 V2 retrievals over that region would not have any surface enhancements and this leads to errors in AMF.

To investigate these ratios and their distribution further, we plotted scatter diagrams of the NASA and ECCC AMFs

and a scatter diagram of NASA and ECCC ratios for all sources (**Figure 4**). On average, ECCC ratios are smaller than NASA ratios: the average values for all sources are 2.2 and 3.8, respectively. For sources located above 2000 m Above Sea Level (ASL), the AMFs for conditions with and without snow are not very different, and there are no large differences between NASA and ECCC AMFs. The snow to snow-free AMF ratios for such sources are not very large; all are between 1.3 and 2.2 for both NASA and ECCC data sets. NASA and ECCC AMFs for snow conditions are not very different from the snow-free

AMFs with majority of the values between 1 and 2. The largest difference between NASA and ECCC AMFs for no-snow conditions can be seen for sources located below 300 m ASL (the light green diamonds in **Figure 4**). While ECCC AMFs for these sources are between 0.2 and 0.4, the NASA AMFs range from 0.2 to 1.3. This difference probably occurs because no-snow, clear sky NASA AMFs are more sensitive to the choice of a priori profiles for low altitude sources. The two largest NASA AMFs belong to the sources in northern Russia mentioned above where a priori profiles are unreliable.

Thus, there are some difference between mean ECCC and NASA AMFs that are largely caused by different assumptions regarding a priori $SO_2$ profile shapes. The ECCC AMF assumptions are more suitable for emission estimates since they better represent the $SO_2$ profile within a plume, while the NASA AMF assumptions are more suitable for average conditions. Since this study is focused on emissions, the presented results are based on ECCC AMFs, except for some case study illustrations, where results for both AMFs are shown to further demonstrate the differences in estimated emissions.

**3.2 Case studies**

**Figure 5** shows four examples of $SO_2$ annual emissions time series estimated for snow and snow-free conditions using NASA and ECCC VCDs derived from OMI and TROPOMI data. The green lines represent emission estimates for snow-free conditions, i.e., the same data as in the Version 2 $SO_2$ catalogue. The cyan lines show emissions estimated using data with snow on the ground, i.e., data that were not used in the $SO_2$ catalogue. There are also two estimates of emissions based on all

data. The red line is the average of the estimates for conditions with and without snow weighted using an inverse-variance weighting method. This approach can be considered as a quick correction of the existing $SO_2$ catalogue when emissions are estimated using only pixels with snow and then those emission estimates are merged with the existing "snow free" catalogue values. The blue line represents the emission estimates based on all data (with and without snow on the ground).



Kliuchevskoi is a volcanic source in Kamchatka, where observations over snow account for 35% of all measurements
suitable for emission estimates. Volcanic degassing emissions are variable, nevertheless both snow and snow-free data show
a similar variability except, perhaps in 2018, when emissions under snow conditions were about 5 times larger than under no-
snow  (**Figure 5 a-c**).  This could be explained by high emission levels in the winter-spring time. TROPOMI data do not show
the same high emission values in 2018; however, no TROPOMI data were available from January to mid-April. This example
also shows that the weighted average of emission estimates under snow and no-snow conditions may not reflect the total annual
emissions correctly. The no-snow estimates show low emissions with small spread, while estimates for snow conditions were,
probably, highly variable with large standard deviations and therefore had a low weight. Emission estimates based on all data
provide a value close to the mean of emissions under snow and no-snow conditions. As Kliuchevskoi is about 5000 m high,
the ECCC and NASA AMF values as well as emission estimates are very similar for these two data versions.

Karabash is a copper smelter in the Ural region of Russia, where observations over snow accounts for 31% of all
measurements. It is expected that smelter emissions would not change very much with the season. **Figure 5e** confirms this, as
the ECCC-AMF-based estimates with and without snow are almost identical in most years.  The standard deviation of the
difference between these estimates is only 14 kt $y^{-1}$ (about 12%). Estimates based on NASA data demonstrate larger (even
twice larger in 2011) emissions when data for snow conditions are used.

East Lambeishor is an oil treatment plant in northern Russia that processes  fluid mixtures of oil, gas, and water from
oil wells, removes hydrogen sulfide, and prepares the oil for further use. It is one of the two sources mentioned in Section 2.1
where the difference between NASA and ECCC ratios is particularly large. The plant started its operation in about 2014, and
therefore its emissions were not properly reflected in the inventories used to derive model $SO_2$ profiles for the NASA retrievals.
NASA AMFs for snow-free conditions are almost the same as for snow conditions (1.3 and 1.8, respectively), and the ratio is
only 1.4, while it is 5 for ECCC AMFs. Snow and snow-free conditions each account for half of the total number of
measurements. **Figure 5h** shows that ECCC estimates for snow and snow-free conditions are similar, while NASA estimates
for snow conditions in 2015-2020 are 2.5 or more times larger than that for snow-free conditions (**Figure 5g**). The ECCC
AMF-based estimates show similar emissions for both conditions for most years. The emission uncertainties for snow-free
conditions are much larger than for snow conditions. In 2019, the emissions for snow-free conditions were about zero, although
the uncertainty of this estimate was large, about 50 kt $y^{-1}$. The 2019 weighted average (the red line in **Figure 5h**) is determined
by the emission value for snow conditions and nearly identical to the estimated annual emissions in the three previous years.
TROPOMI data also show that emissions for snow-free conditions in 2019 are about 80 kt $y^{-1}$, i.e., very close to OMI estimated
for snow conditions. This example demonstrates that additional measurements under snow conditions can substantially
improve the satellite annual emission inventory.

The proper way to validate the estimated emissions for snow and snow-free conditions is to compare them with
reliable inventories of reported point-source emissions. Such inventories are available for the USA and Europe, but as **Figure
1** shows, the number of measurements for snow conditions there is too small for validation. There is, however, information
about annual emissions for Canadian and some Russian sources. Three examples are given below.



The Angarsk source (**Figure 5 j-l**) represents a cluster of two coal-fired power plants (Thermal Power Stations 9 and 10) located 8 km apart in the Irkutsk region of Russia. These plants provide steam, heat, and electricity, and it is expected that

their emissions are higher during the cold season when the plant outputs are used to heat buildings in local residential and industrial areas. As **Figure 5 j, k** shows, emissions for snow conditions are 2-5 times greater than those for snow-free conditions for both ECCC and NASA AMF-based data sets. Uncertainties and inter annual variations in emissions estimates are large for this source. It is interesting to note that the total emissions are rather stable. This example illustrates that there are sources with very large seasonal variations of emissions. Note that there are also other industrial sources in the region, e.g.,

the Angarsk oil refinery, that also contribute to total emissions from the Angarsk source. TROPOMI data show (**Figure 5 l**), in general, total emissions that are similar to those from OMI, although the spread between snow and no-snow estimates is not as large as for OMI. The average 2005-2022 estimated Angarsk $SO_2$ emissions are 122, 31, and 66 kt $y^{-1}$ for snow, snow-free, and all conditions respectively (for ECCC AMFs). The reported annual total emissions from the two power plants were available for 2018-2020 from the Reports on the State and protection of the environment in the Irkutsk Region (in Russian,

available online from https://www.ecoindustry.ru/gosdoklad/view/570.html accessed on November 23, 2022). They were from 75 to 97 kt $y^{-1}$. Thus, emissions based on OMI snow-free data (on average, 33 kt $y^{-1}$ in 2018-2020) are largely underestimated, and additional data for the snow conditions, that account for 35% of all measurements, produce more realistic annual emission estimates (on average, 72 kt $y^{-1}$ in 2018-2020).

Norilsk smelters are the world's largest industrial point source, and column $SO_2$ values over Norilsk are high (**Figure**

**2**). High Norilsk $SO_2$ VCDs are clearly seen by satellite instruments working in the IR-and UV-spectral intervals (Bauduin et al., 2014; Fioletov et al., 2013; Li et al., 2020c; Walter et al., 2012). Norilsk is located in Central Siberia at 69.36° N (above the Arctic Circle). The polar night there lasts 45 days, and observations with SZA<70° are available only from March to September. The mean annual temperature is around −10°C with only 84 days a year with the temperature above 0°C. The snow cover on average lasts for 247 days (Shiklomanov and Laruelle, 2017), and only three summer months are typically snow-free.

Measurements over snow account for 60% of all suitable measurements. Thus, the annual emission estimates in the $SO_2$ catalogue are actually based on just 3-4 months of data and the addition of snow-covered days to the emission estimates doubles that length.

Norilsk emissions are published annually in the Reports on the state and protection of the environment in the Krasnoyarsk Region (in Russian, available online from http://krasecology.ru (e.g.,

http://krasecology.ru/Data/Docs/Сводный%20Доклад%20-%202021.pdf accessed on November 23, 2022)). Reported and estimated emissions for Norilsk are shown in **Figure 6 a**. For the ECCC AMF-based data, the average difference between estimated and reported emissions is -3%, 0%, and -6.4% for all conditions, snow-free and snow conditions respectively, while the standard deviations are 9.3%, 10.3%, and 15.0% respectively. Estimates based on all data are typically closer to reported emissions compared with estimates based on either data with or without snow (**Figure 6a**). The biases between reported and

estimated emissions are within the 2-σ uncertainties and are not unexpected since several factors could produce a multiplicative bias in the estimated emissions (Fioletov et al., 2016).





Flin Flon copper and zinc smelter was one of the largest $SO_2$ emission sources in North America, although its emissions were nearly 10 times less than the Norilsk emissions. The Canadian annual point-source $SO_2$ emission data are available from the National Pollutant Release Inventory (NPRI, https://www.canada.ca/en/services/environment/pollution-

waste-management/national-pollutant-release-inventory.html accessed on December 05, 2022). Smelter operations ceased on June 11, 2010. Flin Flon is located at 54.77° N and typically has snow cover from November to April and a snow-free surface in May-September; therefore, about 37% of all OMI data over Flin Flon suitable for emission estimates are from snow conditions. Our estimated emissions for Flin Flon are typically lower than the reported ones. In the case of all data and ECCC AMFs, the difference is about 15% (**Figure 6b**). The difference is even larger (about 20%) if only snow-free data are used.

Nevertheless, all estimated emissions show high values in 2005-2009, a steep decline in 2010, and nearly zero values thereafter.

## 3.3 Overall statistics

Annual emissions were estimated for conditions with and without snow, and these estimates were compared with the results summarized in **Table 1**. First, the mean annual emissions for the two types of conditions (snow and snow-free) were calculated for every source, and then these mean values were averaged among all sources within that source category (power plant,

smelter together with oil and gas, volcano). Only sources with mean annual emissions exceeding 20 kt y$^{-1}$ over the 2005-2022 period for both snow and snow-free conditions were used to avoid the impact of small sources with high emission uncertainties. Finally, the percent difference between the mean values for conditions with and without snow were calculated for each group. The calculations were done using ECCC AMF-based data.

As **Table 1** shows, there is a 3%-4% average difference (estimates for the snow conditions are larger) between snow

and snow-free conditions for smelters and oil and gas-related sources as well as for volcanos. This difference is relatively small taking into account that the estimates are based on two independent subsets of data and is within its statistical uncertainty. As for individual sources, for three quarters of smelters and oil and gas-related sources, the difference is within ±20%. For power plants, however, the average difference, at about 25%, is much larger than for other types of sources, i.e., power plants have additional 20% difference between snow and snow-free conditions compared to, e.g., smelters. This is consistent with the fact

that most of them serve as a source of heat during the cold season and/or responding to a higher energy demand during the cold season and therefore produce more emissions during that time than during the warm season. The largest difference between the snow and snow-free-based estimates occurs at Angarsk discussed in Section 3.2.

The overall impact of additional measurements for snow conditions on estimated total annual emissions for four main regions is shown in **Figure 7**. The largest differences between the current $SO_2$ catalogue (i.e., for snow-free conditions) and

the emission estimates that include observations over snow can be seen for power plants in the region of Russia plus Kazakhstan. As discussed, it is likely due to larger power generation during the cold season. The difference is particularly large for the thermoelectrical power station Angarsk (see section 3.2). Overall the annual power plants emissions in that region are 15% higher (13% without Angarsk) if measurements over snow are included. In contrast, smelters and oil and gas-related sources in that region show only a 2% difference in estimated emissions when data for snow conditions are added, although





the fraction of measurements over snow is nearly the same for both cases (about 30%). The standard deviation of the difference is also nearly the same in both cases (about 6%). The total emissions from five Ukrainian power plants are also higher, although the mean difference is only 6% due to warmer winter conditions as illustrated by the total number of days with snow on the ground (10%). All sources in China with 10% or more observations over snow are in the north-western part of the country. Although most of them are power plants, the addition of data with snow on the ground practically does not change the average emission estimates (the difference is only 1.5%). For Canada, the mean difference is about 6%, and it is largely caused by the difference for two major sources: Flin Flon and Thompson smelters, that were shut down in 2010 and 2018, respectively. Aside from some biases, the estimates based on snow-free data and all data are consistent: The standard deviation of the difference between annual emissions for them is between 6% and 12% for all regions.

All the emissions estimates above were based on a constant decay time, $\tau = 6$ h. This value was previously found statistically from measurements over snow-free conditions (Fioletov et al., 2015), but we used it for snow conditions as well. The chemical $SO_2$ lifetime is much longer, 13-19 h in summer and about 50 h in winter according to in situ measurements and model calculations (Lee et al., 2011). This suggests that the used decay time $\tau = 6$ h is more related to the plume dispersion that brings $SO_2$ VCDs below the instrument sensitivity level, rather than to the $SO_2$ chemical destruction. If the dispersion is not very different in winter and summer, then the decay times could be also similar in the two seasons. Or, perhaps, the lifetime in the plume is substantially different from that under background conditions. It is difficult to obtain $\tau$ directly from the satellite $SO_2$ VCD data fitting, because the number of sources with snow conditions and the number of measurements for snow conditions is rather limited. Norilsk is probably the only place where such estimates can be reliably done where all three parameters of the fitting (see Fioletov et al., 2015) are estimated from the data. The calculations for Norilsk show $\tau$ values of 8.4 h ±0.5 h and 7.5 h ±0.7 h (2-σ intervals) for snow and snow-free conditions respectively, i.e., different by about 10%. As $\tau$ is a parameter of the fitting function, the overall impact of this difference in the $\tau$ value to the estimated emission is even less; the emissions for snow and snow-free calculated with these $\tau$ values are different by only about 3.5%, and they are different by about 6% from the emission values calculated for the $\tau = 6$ h value using the standard $SO_2$ catalogue one-parameter algorithm. Estimates of $\tau$ using TROPOMI data show similar results: $\tau = 7.4 \pm 1.3$ h for snow and $6.2 \pm 0.6$ h for snow-free conditions in 2018-2022. This shows that an approximation of 6 h for the $SO_2$ lifetime is reasonable and ensures consistency throughout the emission estimates.

## 4.     Summary and discussion

In this study we investigated how satellite $SO_2$ measurements over snow-covered surfaces can be used to improve the "top down" emissions reported in the snow-free $SO_2$ catalogue. OMI data for 2005-2022 were used in the study and the same emission estimation algorithm as in the $SO_2$ catalogue (Fioletov et al., 2016, 2023) was applied. Only 100 out of 759 sources, listed in the catalogue, have 10% or more of the observations over snow. However, for 40 sources, more than 30% of measurements suitable for emission calculations were takes over snow-covered surfaces. The addition of data for snow





conditions is particularly important at high-latitudinal sites. For example, in the case of Norilsk smelters, the world's largest SO$_2$ point source, annual emission estimates in the original SO$_2$ catalogue were based only on 3-4 summer months; additional data for snow conditions extends that period to 7 months.

The SO$_2$ catalogue was based on satellite measurements of SO$_2$ SCDs that were converted to VCDs using single site-specific constant ECCC AMFs, calculated for snow-free conditions. The same approach was applied to measurements with snow on the ground; a new set of constant site-specific AMFs was created and applied to SCDs to obtain VCDs for snow conditions. Then, the emissions were estimated using these VCDs for snow conditions as well as using a merged dataset of VCDs for snow and snow-free conditions. It was found that emissions estimated for snow conditions using ECCC AMFs on

average are consistent with estimated for snow-free conditions for volcanic sources, smelters, and oil and gas-related sources (there is a small, 3-4% bias between them). For power plants, emissions estimated for snow conditions are on average 25% higher than for snow-free conditions. This is likely because many power plants are also used as a heat source in the wintertime.

Similar emissions estimates based on NASA OMI V2 VCD data show larger biases snow and snow-free conditions. Biases are about 20% for volcanic sources, 38% for smelter and oil and gas sources, and more than 60% for power plants.

Analysis of the AMF factors (calculated as a median ratio of SCD to VCD) suggests that the difference between such NASA AMFs for snow and snow-free conditions is typically 1.5-2 times less than that for ECCC AMFs. The reason for that is relatively high NASA AMFs for some sources at low elevation under snow-free conditions that can be explained by non-optimal a priori profiles in the retrievals.

The approach to establish a priori SO$_2$ profiles in the NASA dataset is based on a climatology from CTM model

output may not be optimal for emissions estimates, particularly over the areas around isolated point sources. It is important that the AMFs represent the vertical profile in the plumes correctly in order for the emission estimation algorithms to be accurate. For the NASA dataset, cases are combined when SO$_2$ is in the plume and then it is outside the plume, i.e., without SO$_2$ in the boundary layer. Therefore, the vertical profile represents the "average" conditions that may not describe the vertical distribution in the plume correctly. This effect is particularly important in the areas where SO$_2$ emission sources are not

properly accounted for in the CTM model or in the areas where the spatial resolution of a global model does not sufficiently resolve the gradient in SO$_2$ in the boundary layer (e.g., around major point sources). Examples of oil and gas sources in northern Russia suggest that AMFs created from incorrect a priori SO$_2$ profiles can lead to unrealistic emission estimates (e.g., can be overestimated by a factor of four). On the other hand, if the SO$_2$ is in the free troposphere or stratosphere, e.g., for emissions from volcanic eruptions, an incorrect a priori profile would underestimate volcanic VCDs. The better approach is to make

assumptions about the vertical profile based on the source of SO$_2$, e.g., from local pollutions or from remote volcanic eruptions, to calculate the final retrieved SO$_2$ VCD value.

In summary, is it worth adding data in snow-covered conditions for the annual emission estimates. If we use just two ECCC AMFs (one for snow, the other for snow-free conditions), these AMFs always have some uncertainties due to uncertainties in the input parameters. Moreover, the snow albedo changes depending on snow depth and age. Thus, there could

be some systematic differences between SO$_2$ values (and emissions) retrieved over snow and snow-free conditions.



Nevertheless, as shown for individual sources, the difference between emissions estimated for snow and snow-free conditions is within ±20% for three quarters of smelters and oil and gas sources. This is a reasonable agreement given that there is typically a 3-5 times difference between AMFs for conditions with and without snow (**Figure 3 c**). With just a few exceptions, such as Norilsk, the snow-covered observation conditions represent less than 40% of all observations for typical sites with snow cover

suitable for emission estimates. Therefore, adding snow conditions introduces the bias that is typically less than 20% × 0.4 =8% that is not very large compared to the overall uncertainties of OMI-based annual emissions estimates that are about 50% (Fioletov et al., 2016). We also saw that the difference between winter and summer emissions could be as large as 100% for some power plants. Thus, adding observations with snow-covered conditions could be important to account for seasonal changes in emissions. We are planning to include data for snow conditions for emission estimates in the next version of the

SO$_2$ catalogue. The emission estimates could be further improved in the future if more accurate AMF estimates for different albedo conditions and better estimates of winter and summer decay times become available.

## 5.      Data availability

The OMI version 2 SO$_2$ data are publicly available from the Goddard Earth Sciences (GES) Data and Information Services Center (DISC) (https://disc.gsfc.nasa.gov/datasets/OMSO2_003/summary accessed on June 1, 2023 (Li et al., 2020a). The

TROPOMI COBRA SO$_2$ data set is presently available from available from BIRA-IASB through distributions.aeronomie.be (accessed on April 19, 2023) and will be available from the S5P-PAL Data Porta (https://data-portal.s5p-pal.com/ accessed on June 1, 2023. The version 2 SO$_2$ point source catalogue is available from the NASA Global Sulfur Dioxide Monitoring Home Page (https://so2.gsfc.nasa.gov/measures.html   accessed on April 19, 2023). The direct link to the data set is https://so2.gsfc.nasa.gov/kml/Catalogue_SO2_2022.xls.                    DOI                    identifier

https://disc.gsfc.nasa.gov/datasets/MSAQSO2L4_2/summary  (DOI: 10.5067/MEASURES/SO2/DATA406).

**Author Contributions.** VF prepared the paper and figures with contributions from all the co-authors. VF and CM developed the emission estimation algorithm. CM calculated the AMF factors. VF, CM and DG processed satellite data and estimated the emissions. CL, NK, and JJ developed the OMI PCA V2 algorithms and provided data. NT developed the TROPOMI COBRA algorithms and provided data. SC contributed to cataloguing of volcanic sources and interpreted the estimated

volcanic emissions.

**Competing interests.** The authors declare that they have no conflict of interest.



## Acknowledgments

Can Li, Nickolay Krotkov, and Joanna Joiner acknowledge support from the NASA Earth Science Division Aura Science Team and US participating investigator programs. OMI PCA $SO_2$ data used in this study have been publicly released as part of the Aura OMI Sulfur Dioxide Data Product - OMSO2. We thank EU/ESA/KNMI/DLR for providing the TROPOMI/S5P Level 1 products. Nicolas Theys acknowledges support from ESA (S5P-PAL and S5P MPC projects) and BELSO Prodex-Trace S5P.



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





**Table 1**. The average and standard deviation of the difference (%) between emissions estimates based on OMI measurements for snow and snow-free conditions calculated using ECCC AMFs. The difference between the mean emissions for the period 2005-2021 was calculated first for each source and then the average of these differences was calculated. The total number of sources is also shown. Only sources with mean emissions exceeding 20 kt y$^{-1}$ for both snow and snow-free conditions were included.


| Source Type | Number of sources | Average difference (%) | Standard deviation of the difference (%) |
|---|---|---|---|
| Power plants | 27 | 25.8 | 33.3 |
| Smelters and Oil and Gas | 34 | 4.2 | 20.6 |
| Volcanos | 7 | 3.2 | 27.3 |



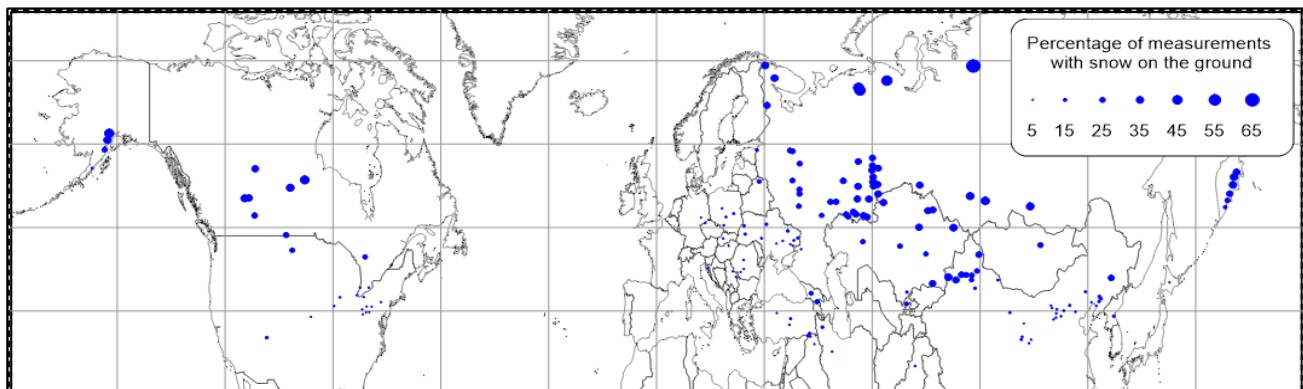

**Figure 1.** Emission sources detected with OMI observations over snow or ice. The map shows the sites and the fraction of OMI satellite pixels that are suitable for emission estimates (with SZA<70 and CF<0.3) for these sites as a percentage of all satellite pixels suitable for such estimates. The symbol size is proportional to that fraction.







**Figure 2.** Mean OMI SO$_2$ slant (a, b) and vertical (c, d) column density over Norilsk for snow-free (a, c) and snow (b, d) conditions for the period 2005-2022. ECCC AMFs were used for the VCD calculations. Data are smoothened by the oversampling technique with a 25 km averaging radius. Slant column values for snow conditions are about 7 times greater than those for snow-free conditions.





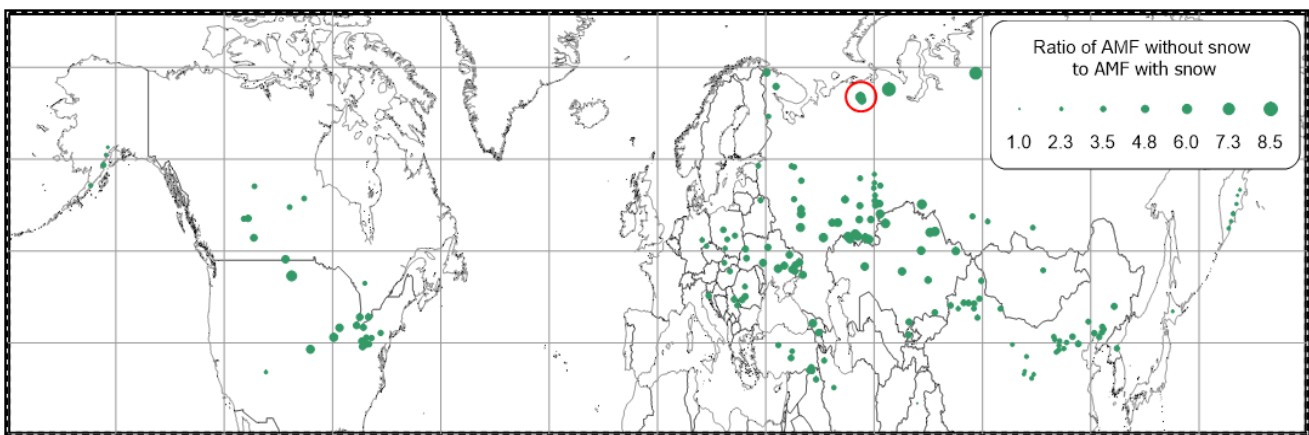


**Figure 3**. The ratio between AMFs for snow condition to AMFs for snow-free conditions for ECCC AMFs. Two recently built sources that are not included in the inventories and therefore do not have reliable NASA AMFs are marked by the red circle.



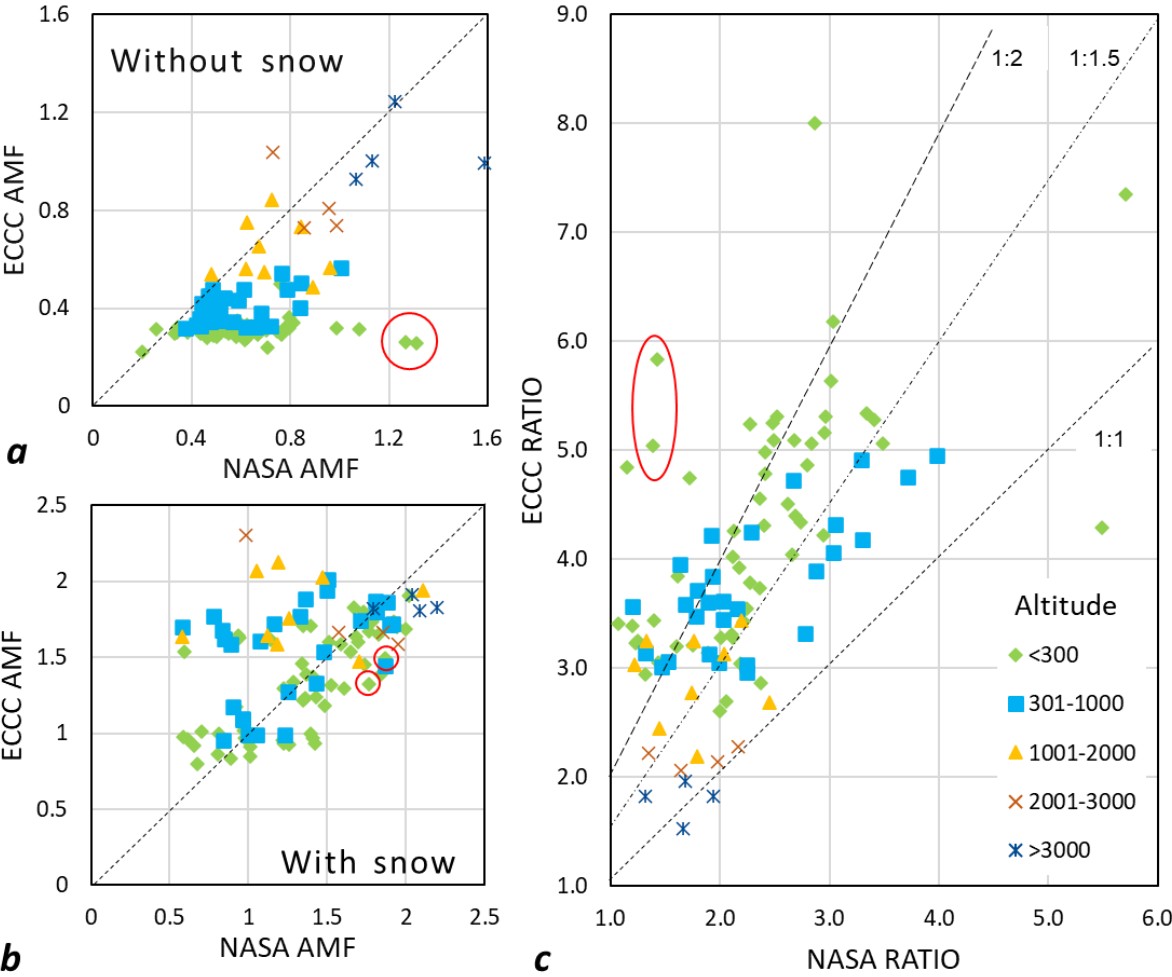

**Figure 4.** Scatter plots of NASA and ECCC AMFs for conditions (a) without and (b) with snow on the ground. (c) Scatter plot of the AMF enhancement due to snow (the ratio between AMF for conditions with snow to that without snow) for NASA and ECCC AMFs. Different colors represent different elevations of the emission sources (in m) as shown in the legend. Two recently built sources that are not included in the inventories and therefore do not have reliable NASA AMFs are marked by the red circles and ellipses.




**Figure 5.** Time series of estimated annual emissions from OMI NASA dataset (the left column), OMI ECCC AMF-based dataset (the middle column) and TROPOMI ECCC AMF-based dataset (the right column). The lines represent estimates based on all data (blue), on data for snow-free conditions (green), on data for snow conditions (cyan), and the weighted average of the two latter datasets (red) with 2-σ error bars. Note that TROPOMI records started in April 2018 and 2018 estimates over snow are not comparable with OMI estimates for that year.






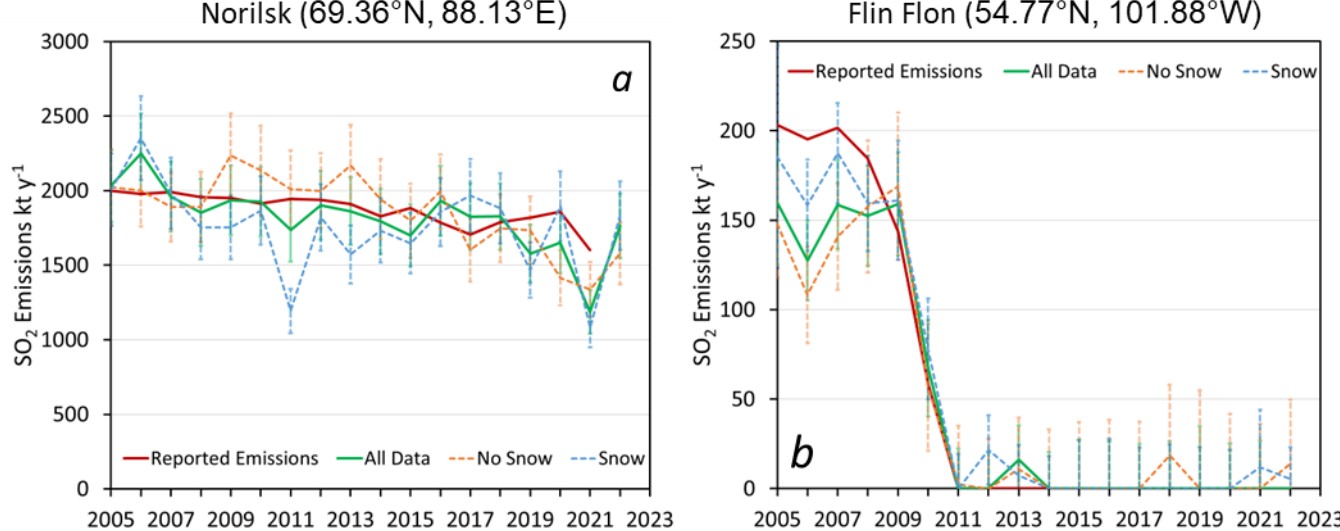

**Figure 6.** Reported and OMI-based estimated SO$_2$ emissions for Norilsk (a) and Flin Flon (b). Reported emissions (solid dark red line) and emissions estimated by applying ECCC AMFs and using all data (solid green line), only pixels without (dashed orange line) and with (dashed blue line) snow on the ground. The error bars represent 2-σ uncertainties.






**Figure 7.** Total emissions for different regions and source types estimated using (blue line) all data (i.e., with and without snow on the ground) and (orange line) only snow-free data. The mean difference between the two estimates, standard deviation of the difference (sigma), and the fraction of satellite measurements taken over snow-covered surface are also shown. Only emission sources where at least 10% of total measurements were taken over snow are included in this analysis. Estimates based on OMI data are shown.