# Peer review of "Estimation of anthropogenic and volcanic SO2 emissions from satellite data in the presence of snow/ice on the ground"

_EGUsphere, 2023_

## Author Comment (AC1)

This study focuses on improving SO2 top-down emission inventory over snow-covered surface. The snow-covered Vertical Column Densities (VCDs) are involved in SO2 emission estimation process for the first time. By creating new AMFs and then VCDs under snow-covered conditions, SO2 emissions of snow-impacted sources can be reevaluated, as more measurements in winter season become available. As a major anthropogenic source of SO2, emissions from power plants in high-latitude regions have shown significant improvement. I recommend publication after some minor corrections mentioned below.

We would like to thank the reviewer for his favorable comment.

At the end of section 2.3, I suggest to place the extension of SO2 observations by including snow-covered pixels in a broader context by also referring to the NO2 product of OMI and TROPOMI, which is already including snow-covered pixels based on similar principles (see the TROPOMI ATBD of NO2, and Van der A et al., 2020, https://doi.org/10.1038/s41612-020-0119-z)

Thank you for your comment. We added references to the papers that discuss OMI and TROPOMI snow-covered pixels in the NO2 data products.

Line 66: "…and volcanic SO2 is is used …". An "is" too many.

Corrected

Line 141: An 10% empirical correction is applied to the OMI VCDs. Is this a positive (+10%) or negative correction (-10%)?

It is a positive (+10%) correction. We added this information.

Line 246: "…norther Russia…" should be "… northern Russia…"

Corrected

Line 457: "In summary, is it worth…" should be "In summary, it is worth…"

Corrected

---

## Author Comment (AC2)

The manuscript " Estimation of anthropogenic and volcanic SO2 emissions from satellite data in the presence of snow/ice on the ground" from Fioletov et al. focusses on improving existing SO2 emission inventories based on satellite data by explicitly including measurements over snow, which are usually excluded in these inventories. These inventories therefore likely underestimate emissions of sources at high latitudes, which are covered by snow for part of a year.

The authors have generated new site-specific AMFs for snow conditions and evaluated the impact on the SO2 inventory, showing a significant improvement.

I recommend publication after some minor corrections:

We would like to thank the reviewer for his favorable comment.

Line 64: " …for assessment the efficiency… " - it should be " …for assessment OF the efficiency…"

Corrected

Line 66: "… volcanic SO2 is is used …" - it should be "… volcanic SO2 is used…"

Corrected

Section 2.3, Line 172ff: I suggest to move your discussion about the SO2 lifetime from Sect. 3.3 line 404ff here, since it gives a very good explanation why you use the same decay time for snow and snow-free conditions and it fits better in this section…

We moved the section and deleted one sentence to make it shorter.

Line 193: What is the reason for using CRF<0.3 for cloud&snow-free conditions? 0.3 is really high and you will certainly use pixels which are partly covered by clouds, especially for the big pixel size of OMI… What happens if you use e.g. CRF<0.1?

We tested other limits (CRF<0.1 and CRF<0.2) for the catalogue paper and found that the emission estimates are very similar, although their uncertainties are somewhat larger than those for CRF<0.3. The limit CRF<0.3 gives slightly more pixels that is important to OMI. Note that the first and last 10 cross-track positions (the largest OMI pixels) were excluded from our analysis.

Line 202ff: I a msising information here which Snow/Ice information for TROPOMI data you use. You describe it for OMI, but not for TROPOMI

The snow cover information was obtained from the Interactive Multisensor Snow and Ice Mapping System measurements (IMS) (Helfrich et al., 2007). It was used for both OMI and TROPOMI. We added this to the text in section 2.4.

Line 202ff: TROPOMI data also contains VIIRS cloud data, that you could use and that provides more accurate results on CF...

Thank you for this information. We will test VIIRS data from TROPOMI files in the future. The present study is focused on OMI data and TROPOMI data are used mostly for illustrative purposes.